# Association between Uremic Toxin Concentrations and Bone Mineral Density after Kidney Transplantation

**DOI:** 10.3390/toxins12110715

**Published:** 2020-11-13

**Authors:** Benjamin Batteux, Sandra Bodeau, Camille André, Anne-Sophie Hurtel-Lemaire, Valérie Gras-Champel, Isabelle Desailly-Henry, Kamel Masmoudi, Youssef Bennis, Ziad A. Massy, Saïd Kamel, Gabriel Choukroun, Sophie Liabeuf

**Affiliations:** 1Department of Pharmacology, Amiens University Medical Center, F-80000 Amiens, France; bodeau.sandra@chu-amiens.fr (S.B.); andre.camille@chu-amiens.fr (C.A.); hurtel-lemaire.anne-sophie@chu-amiens.fr (A.-S.H.-L.); gras.valerie@chu-amiens.fr (V.G.-C.); masmoudi.kamel@chu-amiens.fr (K.M.); bennis.youssef@chu-amiens.fr (Y.B.); liabeuf.sophie@chu-amiens.fr (S.L.); 2MP3CV Laboratory, EA7517, Jules Verne University of Picardie, F-80000 Amiens, France; kamel.said@chu-amiens.fr (S.K.); choukroun.gabriel@chu-amiens.fr (G.C.); 3Department of Rheumatology, Saint-Quentin Medical Center, F-02321 Amiens, France; 4RECIF, Amiens University Medical Center, F-80000 Amiens, France; 5Department of Rheumatology, Amiens University Medical Center, F-80000 Amiens, France; DesaillyHenry.Isabelle@chu-amiens.fr; 6Department of Nephrology, Ambroise Paré University Hospital, APHP, Boulogne Billancourt, F-92100 Paris, France; ziad.massy@aphp.fr; 7Centre for Research in Epidemiology and Population Health (CESP), INSERM UMRS 1018, Université Paris-Saclay, F-94807 Villejuif, France; 8Biochemistry Laboratory, Amiens University Medical Center, F-80000 Amiens, France; 9Department of Nephrology Internal Medicine Dialysis Transplantation, Amiens University Medical Center, F-80000 Amiens, France

**Keywords:** uremic toxin, bone mineral density, fracture, kidney transplantation

## Abstract

Although uremic osteoporosis is a component of mineral and bone disorder in chronic kidney disease, uremic toxin (UT) concentrations in patients with end-stage kidney disease and bone mineral density (BMD) changes after kidney transplantation have not previously been described. We hypothesized that elevated UT concentrations at the time of transplantation could have a negative impact on bone during the early post-transplantation period. Hence, we sought to determine whether concentrations of UTs (trimethylamine-N-oxide, indoxylsulfate, p-cresylsulfate, p-cresylglucuronide, indole-3-acetic acid, hippuric acid, and 3-carboxy-4-methyl-5-propyl-furanpropionic acid) upon transplantation are predictive markers for (i) osteoporosis one month after transplantation, and (ii) a BMD decrease and the occurrence of fractures 12 and 24 months after kidney transplantation. Between 2012 and 2018, 310 kidney transplant recipients were included, and dual-energy X-ray absorptiometry was performed 1, 12, and 24 months after transplantation. The UT concentrations upon transplantation were determined by reverse-phase high-performance liquid chromatography. Indoxylsulfate concentrations upon transplantation were positively correlated with BMD one month after transplantation for the femoral neck but were not associated with osteoporosis status upon transplantation. Concentrations of the other UTs upon transplantation were not associated with osteoporosis or BMD one month after transplantation. None of the UT concentrations were associated with BMD changes and the occurrence of osteoporotic fractures 12 and 24 months after transplantation. Hence, UT concentrations at the time of kidney transplantation were not predictive markers of osteoporosis or fractures.

## 1. Introduction

Chronic kidney disease-mineral and bone disorder (CKD-MBD) is characterized by one or more of the following manifestations: (i) renal osteodystrophy (ROD), (ii) vascular and soft tissue calcification, and (iii) abnormal metabolism of calcium, phosphorus, parathyroid hormone (PTH) or vitamin D [1]. In uremic patients, ROD comprises histologically evidenced abnormalities in bone turnover, mineralization, volume, linear growth, and strength [2].

Mild hyperparathyroidism-related bone disease (histologically reflected by high-turnover bone, which leads to osteitis fibrosa in advanced cases) is more frequently encountered in early-stage kidney disease [3]. Adynamic bone disease (ABD, characterized by reduced osteoblasts and osteoclasts, no accumulation of osteoid and markedly low bone turnover) [4,5] is also frequently encountered in end-stage kidney disease (ESKD). The histologic pattern of ABD is generally associated with low levels of PTH. In patients with CKD, however, serum PTH levels are generally higher than normal—even when ABD is present. Hence, it is considered that bone tissue is resistant to PTH; a relative reduction in the PTH level is therefore able to induce the emergence of a low-turnover state [5]. Osteomalacia and mixed uremic osteodystrophy can also be encountered in patients with CKD [2]. Regardless of the cause, low bone mineral density (BMD) in patients with CKD is a marker of bone fragility [6]. 

The decrease in BMD after kidney transplantation is generally attributed to the cumulative effect of corticosteroid treatment [7,8,9,10]. Other risk factors include the time on dialysis prior to transplantation, age at transplantation, vitamin D deficiency, and a low body mass index (BMI, <23 kg/m²) [11,12]. Moreover, CKD-associated periodontitis [13] might have a role in bone loss in patients with CKD and in kidney transplant recipients. Various contributory factors (such as pro-inflammatory cytokines [14], oxidative stress [15], matrix metalloproteinases and transglutaminases [16], and asymmetric dimethylarginine [17]) have been identified.

The fracture risk after kidney transplantation appears to be related to a low BMD; indeed, Evenepoel et al. found an association between incident fractures on one hand and BMD at the lumbar spine and at the femoral neck on the other [18]. Furthermore, the latest Kidney Disease Improving Global Outcomes (KDIGO) CKD-MBD guidelines suggest testing for BMD to assess the fracture risk in this population [19,20].

Uremic osteoporosis is a newly recognized component of CKD-MBD [21]. Conventionally, uremic toxins (UTs) are divided into three groups: small molecules (e.g., urea, phosphate, and trimethylamine-N-oxide (TMAO)), medium molecules (e.g. fibroblast growth factor (FGF)-23), and protein-bound molecules (indoxylsulfate (IxS), p-cresylsulfate (pCS), p-cresylglucuronide (pCG), indole-3-acetic acid (IAA), hippuric acid (HA), and 3-carboxy-4-methyl-5-propyl-furanpropionic acid (CMPF)) [22]. In patients with ESKD, UT concentrations are high and somewhat variable [23]. After kidney transplantation, the concentrations of protein-bound UTs fall rapidly to below levels observed in non-transplanted patients with equivalent glomerular filtration rates [24,25,26].

It has already been shown that high levels of IxS, pCS, and IAA are associated with cardiovascular comorbidities and mortality [24,25,27,28]. Regarding bone damage, IxS is involved in skeletal resistance to PTH [29] and inhibits bone resorption [30]. In vitro, pCS induces osteoblast dysfunction by activating the Janus kinase (JNK) and mitogen-activated protein kinase (MAPK) p38 pathways [31]. To the best of our knowledge, there are no published data on the other protein-bound UTs and on TMAO.

In patients with CKD, bone loss is associated with several nonconventional factors (such as immunosuppressive treatments, age, vitamin D deficiency, and low body mass index [11,12]) as well as with conventional osteoporosis risk factors (such as sex, previous fragility fracture, rheumatoid arthritis, parental hip fracture, current cigarette smoking, alcohol intake of ≥3 units per day, hypogonadism, hyperthyroidism, primary hyperparathyroidism, gastrointestinal diseases, Cushing’s syndrome, or idiopathic hypercalciuria [32,33]). Considering the high levels of UTs in pre-transplant patients [23] and the compounds’ bone toxicity in vitro [29,30,31], we hypothesized that UTs would constitute an additional nontraditional risk factor for bone loss in pre-transplant patients. We further hypothesized that elevated UT concentrations upon transplantation would have a negative impact on bone during the early post-transplantation period.

The objective of the present study was therefore to determine whether concentrations of UTs (protein-bound UTs and TMAO) upon transplantation are predictive markers for (i) osteoporosis one month after transplantation, and (ii) a BMD decrease and the occurrence of fractures 12 and 24 months after kidney transplantation.

## 2. Results

### 2.1. Study Population

Of 417 patients having received a kidney transplant in Amiens University Medical Center (Amiens, France) between January 1st, 2012, and June 15th, 2018, 310 (including 194 men, 62.6%) were included in the present study. The mean ± SD age of the study population at the time of transplantation was 51.1 ± 12.8. The most common indication for kidney transplantation was recurrent glomerulonephritis (29.7%). The median (range) time on hemodialysis before transplantation was 2.5 years (0–30.7). The median times to the first, second, and third BMD measurements were 32 days (M1), 12 months (M12), and 24 months (M24), respectively (Table 1).

Data on the lumbar spine BMD were available at M1 and M12 for all 310 patients and at M24 for 222 of the latter. Data on the femoral neck BMD were available at M1 for 224 patients, at M12 for 190, and at M24 for 159. Lastly, total hip BMD data were available at M1 for 278 patients, at M12 for 304, and at M24 for 224.

The median (interquartile range) UT concentrations were 16.1 (9.02–25.60) for pCS, 2.57 (0.97–5.08) for CMPF, 19.80 (12.82–28.32) for IxS, 0.80 (0.25–1.58) for pCG, 25.45 (10.90–50.15) for HA, 4.27 (2.54–7.78) for TMAO, and 0.75 (0.58–1.06) for IAA (Table 1). The concentration distributions are represented in Figure 1.

### 2.2. Uremic Toxin Concentrations upon Transplantation

Upon transplantation, all the UT concentrations were significantly correlated with each other, with the exception of CMPF vs. pCS (rho = 0.11, *p* = 0.053). The strongest correlations were observed for pCG vs. pCS (rho = 0.52, *p* < 0.001), pCG vs. HA (rho = 0.5, *p* < 0.001), and IxS vs. HA (rho = 0.47, *p* < 0.001) (Figure 2, Appendix A).

### 2.3. Correlations between UT Concentrations and BMD at M1 and Changes in BMD at M12 and at M24

Twelve and 24 months after transplantation, the BMD had decreased significantly (vs. M1) at all measurement sites, with the exception of the lumbar spine at M12 (Figure 3, Appendix A).

At the lumbar spine at M1, 184 patients (59.4%) had a normal BMD, 107 (34.5%) had osteopenia, and 19 (6.1%) had osteoporosis (Figure 4). 

Uremic toxin concentrations upon transplantation were not associated with osteoporosis status at M1 (Figure 4) or with BMD at M1 at any of the measurement sites, except for the HA level for the femoral neck (rho = −0.14, *p* = 0.036) and IxS for the femoral neck and the total hip (rho = +0.13, *p* = 0.049 and rho = +0.13, *p* = 0.037, respectively) (Table 2). After adjustment for confounding factors (i.e., sex, BMI, serum calcium, phosphate, 25(OH) vitamin D3, PTH, bone alkaline phosphatases, and osteocalcin levels, Appendix A), IxS concentrations upon transplantation were still positively correlated with BMD changes at the femoral neck at M1 (*p* = 0.042, Appendix A).

Uremic toxin concentrations upon transplantation were not associated with BMD gain at any measurement site 12 months after transplantation (Table 3) or with BMD changes 12 and 24 months after transplantation (Table 4).

The same trends were found after stratification by sex, age group, ABD, kidney failure with a year of transplantation, and early steroid withdrawal (Appendix A).

Four patients experienced an osteoporotic fracture (affecting the L3 vertebrae, the hip, a rib, and the glenoid process of the scapula) 12 and 24 months after transplantation, corresponding to an incidence of 13.5 fractures per 1000 person-years. There was no association between the UT concentrations upon transplantation and the occurrence of osteoporotic fractures within 12 and 24 months of transplantation.

### 2.4. Other Factors Influencing BMD at M1 and Changes in BMD at M12 and M24

BMD at M1 was significantly correlated with (i) BMI for all measurement sites (lumbar spine, ρ = +0.31, *p* < 0.001; femoral neck, ρ = +0.27, *p* < 0.001; total hip, ρ = +0.42, *p* < 0.001), (ii) the serum calcium level (for the lumbar spine only; ρ = −0.12, *p* = 0.036), and (iii) the serum PTH level for the lumbar spine (ρ = −0.17, *p* = 0.003) and the total hip (ρ = −0.13, *p* = 0.040)) (Appendix A). Moreover, BMD at M1 was significantly higher (i) in men than in women for all measurement sites (lumbar spine, 1.034 ± 0.16 vs. 0.970 ± 0.16, respectively; *p* < 0.001; femoral neck, 0.788 ± 0.016 vs. 0.693 ± 0.12, *p* < 0.001; total hip, 0.929 ± 0.15 vs. 0.822 ± 0.13, *p* < 0.001), (ii) in patients with diabetes mellitus than in patients without diabetes mellitus (for the lumbar spine only: 1.130 ± 0.15 vs. 1.007 ± 0.16, respectively; *p* = 0.033) and (iii) in steroid-naive patients than steroid-exposed patients (for the lumbar spine: 1.024 ± 0.16 vs. 0.944 ± 0.015, respectively; *p* < 0.001; and the total hip: 0.899 ± 0.014 vs. 0.852 ± 0.018, *p* = 0.048) (Appendix A). After adjustment, BMD at M1 was (i) positively correlated with BMI for all measurement sites (lumbar spine, *p* < 0.001; femoral neck, *p* = 0.001; total hip, *p* < 0.001), (ii) negatively correlated with the serum calcium level for the lumbar spine (*p* = 0.006) and the total hip (*p* = 0.005), and (iii) negatively correlated with the serum PTH level for the lumbar spine (*p* = 0.014). Moreover, after adjustment, BMD at M1 was significantly higher in men than in women for all measurement sites (lumbar spine, *p* = 0.002; femoral neck, *p* = 0.004; total hip, *p* < 0.001).

BMD changes at M12 (vs. M1) were significantly correlated with (i) age for the total hip (ρ = −0.16, *p* < 0.006), (ii) PTH (for the femoral neck (ρ = +0.20, *p* = 0.019) and the total hip (ρ = +0.19, *p* = 0.002)), (iii) the serum bone alkaline phosphatase level (for the femoral neck (ρ = +0.19, *p* = 0.038) and the total hip (ρ = +0.26, *p* < 0.001)), and (iv) the serum osteocalcin level (for the total hip only (ρ = +0.20, *p* = 0.009) (Appendix A). Moreover, the mean ± SD BMD changes at M12 differed significantly when comparing (i) patients with primary hyperparathyroidism vs. those without (for the lumbar spine (−0.077 ± 0.10 vs. +0.001 ± 0.06, *p* = 0.001, respectively) and the femoral neck (−0.117 ± 0.06 vs. −0.006 ± 0.05, *p* < 0.001, respectively)), (ii) patients with secondary hyperparathyroidism vs. those without (for the femoral neck only: −0.004 ± 0.05 vs. −0.041 ± 0.06, respectively, *p* = 0.005), (iii) patients taking bisphosphonates vs. all the other patients (for the lumbar spine only: +0.054 ± −0.003, *p* = 0.003, respectively), and (iv) patients with early steroid withdrawal after transplantation vs. those on long-term corticosteroid therapy (for the lumbar spine only: +0.031 ± 0.07 vs. −0.006 ± 0.06, respectively, *p* < 0.001) (Appendix A).

BMD changes at M24 (vs. M1) were significantly correlated with (i) serum phosphate (for the lumbar spine (ρ = +0.14, *p* = 0.033) and the femoral neck (ρ = +0.24, *p* = 0.011)), (ii) serum PTH (for the femoral neck (ρ = +0.28, *p* = 0.003) and the total hip (ρ = +0.19, *p* = 0.006)), (iii) serum bone alkaline phosphatases (for the total hip only: ρ = +0.22, *p* = 0.005), (iv) serum osteocalcin for the femoral neck only (ρ = +0.28, *p* = 0.017) and (v) serum creatinine at M12 at all measurement sites (lumbar spine, ρ = −0.13, *p* = 0.048; femoral neck, ρ = −0.28, *p* = 0.003; total hip, ρ = −0.15, *p* = 0.028) (Appendix A). Moreover, the mean ± SD BMD changes at M24 differed significantly in (i) patients with primary hyperparathyroidism vs. those without (for the lumbar spine only: –0.098 ± 0.010 vs. −0.017 ± 0.06, respectively, *p* = 0.003), (ii) patients taking bisphosphonates vs. all the other patients (for the lumbar spine only: +0.053 ± 0.03 vs. –0.023 ± 0.07, respectively, *p* < 0.001), and (iii) patients with early steroid withdrawal after transplantation vs. those on long-term corticosteroid therapy (for the lumbar spine only: +0.036 ± 0.06 vs. −0.024 ± 0.06, respectively, *p* < 0.001) (Appendix A).

## 3. Discussion

Our results provide an overview of BMD as a surrogate marker of bone health and its link with serum UT concentrations in kidney transplant recipients. Our main finding (after adjustment) was the positive association between the IxS concentration upon transplantation and BMD at the femoral neck one month after kidney transplantation (*p* = 0.042). Twelve and 24 months after transplantation, there were no significant associations between UT concentrations upon transplantation and BMD changes. Classic risk factors (such as sex, BMI, and serum PTH, bone alkaline phosphatase, osteocalcin and creatinine levels) appeared to account for the BMD decrease after kidney transplantation.

Primary mineralization disorders are frequent in uremic patients. Indeed, secondary hyperparathyroidism is histologically reflected by high bone turnover, normal mineralization, and low bone volume (depending on the duration of the disease process); it leads to osteitis fibrosis in severe cases. This type of ROD is encountered in early-stage kidney disease [3]. In ESKD, skeletal resistance to PTH’s action [36] is caused by a reduction in osteoblast PTH receptor expression [37]. This “functional hypoparathyroidism” leads to ABD, which corresponds to low-turnover bone with normal mineralization and a normal or low bone volume [4]. Osteomalacia is scarcer in CKD and corresponds to low-turnover bone with abnormal mineralization (an accumulation of osteoid tissue). The bone volume may be low to medium, depending on the severity and duration of the process and the presence of absence of other factors affecting bone [2]. Osteomalacia results from vitamin D deficiency or metabolic acidosis caused by CKD [38]. There is no consensus definition of mixed uremic osteodystrophy (also encountered in patients with CKD) but this is generally described as high-turnover bone with normal bone volume and abnormal mineralization [2]. Regardless of the cause, however, a low BMD in patients with CKD is a marker of bone fragility [6]. According to the latest KDIGO CKD-MBD guidelines, BMD should be measured in patients with stage 3a to 5D CKD with evidence of either MBD, risk factors for osteoporosis, or both, in order to establish whether the fracture risk will impact treatment decisions [19]. Posttransplant CKD-MBD reflects the effects of immunosuppression, the persistence of previous CKD-MBD after transplantation, and de novo CKD-MBD. Our present results showed that (i) respectively, 34.5% and 6.1% of the patients presented osteopenia and osteoporosis one month after kidney transplantation and (ii) 1% of the patients presented a fracture in the 24 months following kidney transplantation. Evenepoel et al. have shown that over a median follow-up period of 5.2 years, 38 out of 518 (7.3%) de novo renal transplant recipients experienced a fragility fracture; this corresponds to an incidence rate of 14.2 fractures per 1000 person-years [18].

Periodontitis can also affect the bone structure and thus must be considered. This multifactorial, chronic, inflammatory disease is caused by periodontal bacteria and is characterized by the progressive destruction of the tooth-supporting apparatus, which includes the alveolar bone [39]. There is a two-way relationship between CKD and periodontitis [40]. On one hand, (i) systemic inflammation (mediated by pro-inflammatory cytokines such as IL-1, IL-6, TNFα, and TGFβ) appears in response to the chronic infection, and (ii) the dissemination of periodontal pathogenic bacteria, the latter’s products, and antibodies developed in response to the infection [41,42] can contribute to a deterioration of renal function via proteinuria [43,44]. On the other hand, (i) the uremic environment due to CKD modulates cytokine and inflammation molecules [45] and (ii) pathological conditions caused by CKD (such as diabetes mellitus [39], poor oral health, and decreased water drinking by hemodialysis patients [46]) can lead to periodontal disease. Moreover, there is probably also a two-way relationship between periodontitis (characterized by alveolar bone resorption) and systemic bone loss [47,48]. Even though osteoporosis is defined as a systemic skeletal disease [32], various cross-sectional and longitudinal studies show that deterioration of the microarchitecture can affect alveolar bone [49]. Furthermore, periodontitis can influence systemic bone density and bone loss through the systemic inflammatory burden mentioned above [50]. Quality of life can also be altered in patients suffering from periodontal disease, which causes tooth loss if not treated early. Advanced periodontitis can lead to a decline in chewing ability, word pronunciation, and aesthetic function [51]. This altered qualify of life particularly affects patients with ESKD, as highlighted by Oliveira et al.’s finding that mild/moderate periodontitis and severe periodontitis were significantly associated with poorer oral-health-related quality of life, relative to the absence of periodontitis (risk ratio (95% confidence interval) = 1.49 (1.16–1.91) and 1.77 (1.36–2.30), respectively) [52]. Thus, periodontal disease must be screened and managed in this population–especially since treatment is associated with a reduction in systemic inflammation [53] and an increase in the glomerular filtration rate [54].

Various research groups have confirmed the relationship between certain UTs and complications of CKD. The UTs studied here were those for which cardiovascular and bone toxicities have been best documented in the literature. Plasma UT concentrations rise as CKD progresses; in patients with ESKD, the values are often more than 10-fold higher than in healthy controls [27,55]. These high levels reflect both increased intestinal production and absorption and decreased renal clearance. Our results confirmed these literature reports, and all the UTs studied here showed the same trends. Furthermore, we evaluated associations between seven UTs. The correlation matrix evidenced weak correlations between the various compounds. Our previous results in hemodialysis patients highlighted differences between UTs with regard to their association with the estimated glomerular filtration rate [56]. At elevated concentrations, UTs may disturb several biological processes and thus become both directly and indirect toxic in various cells and tissues. One mechanism involves the intracellular generation of oxidative stress [57]. Indeed, it has been reported that exposure to several well-known UTs (including inorganic phosphate, pCS, IxS, and FGF23) leads to vascular dysfunction [58].

Hence, most of the research on UTs in patients with CKD has focused on vascular dysfunction. In view of the links between vascular calcification and bone disease, we sought to better characterize the associations between UT levels and bone disease. In vitro, several UTs appear to be harmful for bone. IxS increases Klotho gene hypermethylation, which in turn leads to the progression of vascular calcification. Organic anion transporter 3 (OAT-3, known to mediate the uptake of UTs) is expressed by osteoblasts. IxS is associated with a decrease in PTH receptor expression by osteoblasts. In this model, the OAT-3 inhibitor probenecid decreases IxS-induced skeletal resistance to PTH [29]. As well as suppressing bone formation, IxS inhibits bone resorption by reducing IL-1 and RANKL expression [30]. pCS also induces osteoblast dysfunction by activating JNK and MAPK p38 pathways [31]. In a rat model of kidney failure showing low bone turnover, Iwasaki et al. showed that IxS accumulation was associated with a lower bone formation rate and downregulation of osteoblast-related genes [59]. The animals’ conditions were improved by treatment with the oral adsorbent AST-120—probably due to a reduction in IxS levels.

In contrast to the data from animal studies, Barreto et al.’s clinical study of the association between IxS levels and biochemical parameters related to mineral metabolism and bone histomorphometry in patients with CKD found that the serum IxS concentration was positively associated with the bone formation rate, the osteoid volume osteoblast surface, and the fibrosis volume [60]. The researchers used a multivariate model including intact PTH, FGF-23, ionized calcium, and 1,25-OH-vitamin D_3_ levels [60]. One possible explanation for the apparent conflict between the clinical results and the animal results is that high levels of IxS may affect PTH secretion by participating in skeletal resistance to PTH or decreasing calcitriol synthesis. The animal study was performed with thyroparathyroidectomized uremic rats in which physiological plasma PTH concentrations were restored by infusion [59]; this differed from clinical conditions, where ESKD is most frequently associated with supranormal increases in PTH secretion [5].

Furthermore, we found that IxS concentrations upon transplantation were positively correlated with BMD at M1 for the femoral neck and for the total hip before adjustment (rho = +0.13, *p* = 0.049 and rho = +0.13, *p* = 0.037, respectively) and for the femoral neck after adjustment (*p* = 0.042); these findings corroborate Barreto et al.’s results. After adjustment, however, there was no longer any correlation between IxS concentrations upon transplantation and BMD changes 12 and 24 months after transplantation. This was probably due to the fact that IxS concentrations fall rapidly after transplantation [24] and thus are unlikely to influence the bone mass.

There are several possible explanations for the lack of an association in the present study. Firstly, factors other than UTs (such as sex, BMI, and serum PTH, bone alkaline phosphatase, osteocalcin and creatinine levels, as found in the present study) might better explain the osteoporotic status of kidney transplant receipts. After kidney transplantation, concentrations of protein-bound UTs fall rapidly within a month of transplantation and go below the values seen in nontransplanted patients with equivalent glomerular filtration rates [24,26]. This is why we decided to assay serum UT concentrations upon transplantation, and might account for the lack of correlation between UT concentrations at the time of transplantation and changes in BMD at M12 and at M24. Furthermore, we have recently shown that corticosteroid use in general and exposure time in particular have an impact on BMD and on fracture incidence among kidney transplant recipients [10]. Secondly, the choice of BMD as a surrogate marker of bone health might not be the most appropriate because dual-energy X-ray absorptiometry (DXA) does not differentiate between cortical bone and trabecular bone. The gold standard for the quantitative assessment of bone health is the histomorphometric analysis of a bone biopsy. However, this procedure is invasive. Hence, UTs might alter bone quality but not the BMD. For example, IxS leads to skeletal resistance to PTH [29] (involved in ABD) [36,37] but does not affect BMD [1,2]. Thirdly, we did not have DXA data for the time of transplantation. Nevertheless, we considere that the BMD at M1 post-transplantation was probably similar to the value at M0 because it is known that PTH, phosphate, calcium, calcitriol, and FGF-23 levels take 3–12 months to stabilize [61,62]. Lastly, UT concentrations upon transplantation were high (relative to healthy controls) in our patient population and were highly variable; this variability might have prevented us from detecting a negative impact of UT on bone.

The present study’s strengths included the well-characterized cohort and the accurate, validated UT assay methods. Our study also had several limitations. Firstly, we only studied seven of the dozens of known UTs; however, the protein-bound UTs and TMAO are the most important in terms of toxicity. Secondly, we performed a single-center study, with all its inherent limitations. However, BMD was always evaluated with the same device, and thereby made our intergroup comparisons more robust.

## 4. Conclusions

In contrast to literature data from animal studies showing a potential link between serum UT concentrations and bone abnormalities, the first ever clinical study of this topic failed to show any relationship between concentrations of protein-bound UTs and TMAO upon kidney transplantation and BMD or the occurrence of osteoporotic fractures. Conventional risk factors (such as sex, BMI, and serum PTH, bone alkaline phosphatase, osteocalcin and creatinine levels) appear to account for the BMD decrease after kidney transplantation. Further investigation is required to confirm and explain the present results.

## 5. Materials and Methods

### 5.1. Study Design and Participants

We performed a longitudinal study of a cohort of consecutive dialysis patients (excluding cases of preemptive kidney transplantation) aged 18 and over and having undergone kidney transplantation at Amiens University Medical Center (Amiens, France) between 1 January 2012, and 15 June 2018, and who had available data on BMD measured at several different anatomic sites (the lumbar spine, hip, and femoral neck) 1 month (M1) and 12 months (M12) after transplantation. DXA data were extracted from consultation reports. Bone mineral density was determined using DXA Hologic Discovery System (Hologic Inc., Waltham, MA, USA). Osteoporosis was defined as a T-score ≤ –2.5 at one or more sites, and osteopenia was defined as a T-score between –1 and –2.5 at one or more sites. The study was approved by the French National Data Protection Commission (Commission Nationale de l’Informatique et des Libertés (Paris, France); registration number: PI2019_843_0055) on 18 July 2019. Patients were provided with information about the study, and were free to refuse to participate. In line with the French legislation on noninterventional studies, approval by an investigational review board was neither required nor sought. Upstream, the patients consented to the use of their personal clinical and laboratory data for research purposes.

### 5.2. Collected Data

The variables recorded at baseline included sociodemographic characteristics, osteoporosis risk factors influencing BMD (ethnic group, BMI, thyroid disorders, prior osteoporotic fractures, family history of fracture of the upper extremity of the femur, diabetes mellitus, chronic inflammatory rheumatism, primary hyperparathyroidism, smoking, alcohol consumption, and menopausal status in women), the use of drugs potentially influencing BMD at the time of transplantation, the etiology of chronic kidney disease, the donor’s characteristics, and the characteristics of the transplantation and the immunosuppressive regimens.

Serum levels of calcium, phosphate, calcitriol, PTH, bone alkaline phosphatase, osteocalcin and creatinine, and the glomerular filtration rate at baseline were recorded retrospectively. The serum creatinine level 12 months after transplantation was also recorded. Serum samples were obtained from patients in the 24 hours before transplantation and were stored at −80 °C in the Picardy Biobank (Amiens, France).

Data on prevalent osteoporotic fractures and intakes of calcium, vitamin D and bisphosphonate during the study period were retrospectively collected from medical records.

### 5.3. Identification of Patients with ABD

In order to take into account the heterogeneity of skeletal involvement, patients with PTH <150 pg/mL (positive predictive value for ABD: 97% [63]) and bone alkaline phosphatases <10 ng/mL (which can further bolster the diagnosis of ABD, as it is 100%-sensitive and 93.7%-specific [63]) were classified as having ABD.

### 5.4. Assays of Serum PTH, 25(OH) Vitamin D, Bone Alkaline Phosphatases, and Osteocalcin

Serum PTH was assayed using a chemiluminescent immunoassay (ADVIA Centaur PTH from Siemens Healthcare Diagnostics SAS; intra-assay coefficient of variation <2.05%, inter-assay coefficient of variation <4.04%; detection limit of the assay: 4.6 pg/mL), according to the manufacturer’s instructions. Serum 25(OH) vitamin D was assayed using an ELISA (assay kit total Vitamin D -ADVIA Centaur from Siemens Healthcare Diagnostics SAS). The detection threshold for serum 25(OH) vitamin D was 4.20 ng/mL (intra-assay coefficient of variation <9.79%, inter-assay coefficient of variation <7.32%). Bone alkaline phosphatase and osteocalcin were assayed using Liaison-XL from DiaSorin SA. The detection thresholds for serum bone alkaline phosphatase and osteocalcin were 3.0 µg/L and 1.5 ng/mL, respectively. The intra-assay coefficients of variation were <2.16% and <4.99%, respectively, and the inter-assay coefficients of variation were <4.19% and <5.33%, respectively.

### 5.5. Uremic Toxin Assays

Blood levels of UTs (CMPF, HA, IAA, IS, pCS, pCG, and TMAO) were determined by liquid chromatography (Shimadzu, Marne-la-Vallée, France) coupled to a tandem mass spectrometer (3200 QTRAP, Sciex, Les Ulis, France) [64]. Briefly, UTs were extracted from 50 µL of serum by adding 200 µL of an ice-cold acetonitrile solution containing the internal standards (d5-CMPF, (13C6)-1H-IAA, (13C6)-IxS, d4-pCS, and d9-TMAO) at a concentration of 500 ng/mL. After centrifugation, 50 µL of the supernatant was diluted 20-fold in ultra-pure water (total volume: 1000 µL) before transfer to a vial for injection into the chromatographic system. Chromatographic separation was performed at 40 °C on an ultra PFP propyl column (5 µm, 50 × 2.1 mm, Restek, Lisses, France). The column was eluted with a gradient of acetonitrile with 0.1% formic acid and ultrapure water with 0.1% formic acid at a flow rate of 0.8 mL/min. Data were acquired in multiple reaction monitoring mode after ionization in negative electrospray ionization mode (for CMPF, HA, IxS, pCS, and pCG) or positive electrospray ionization mode (for IAA and TMAO). 

The mean ± SD reference values for subjects with normal kidney function were 6.6 ± 3.7 µg/mL for pCS, 1.1 ± 1.1 µg/mL for CMPF, 0.5 ± 0.3 µg/mL for IxS, under limit of quantification for pCG, 1.3 ± 1.6 µg/mL for HA, 0.5 ± 0.2 for IAA [34], and 3.3 (3.1–6.0) for TMAO [35].

### 5.6. Statistical Analysis

In our descriptive analysis, categorical variables were expressed as the number (percentage), and continuous variables were expressed as the mean ± standard deviation (SD) or the median (range), depending on the data distribution. The Shapiro–Wilk test was used to determine whether or not data were normally distributed. We performed a matrix correlation test on the UT concentrations at transplantation. Correlations between UT concentrations at transplantation and BMD (at the lumbar spine, hip, and femoral neck) 1, 12, and 24 months after kidney transplantation were quantified by calculating Spearman’s coefficient. Uremic toxin concentrations at transplantation were compared for patients with BMD gain vs. those with BMD loss, and for patients with a prevalent osteoporotic fracture vs. those without a fracture. The same analyses were performed after stratification by sex, age group, adynamic bone disease, kidney failure within 12 months of transplantation, and early steroid withdrawal. Correlations between age, BMI, serum levels of calcium, phosphate, calcitriol, PTH, bone alkaline phosphatase, osteocalcin and creatinine (at the time of transplantation—except for creatinine, measured at M12), and (i) BMD at M1 and (ii) changes in BMD at M12 and at M24 (vs. M1) were quantified by calculating Spearman’s coefficient. Student’s t-test was used to assess the statistical significance of associations between binary osteoporosis risk factors and (i) BMD at M1 and (ii) changes in BMD at M12 and at M24 (vs. M1).

In order to consider confounding factors for mineral bone disorders, multiple linear regression models were built for UT concentrations upon transplantation that were significantly correlated with BMD at M1 or with changes in BMD 12 and 24 months after kidney transplantation in bivariate analyses.

All statistical tests were two-sided, and the threshold for statistical significance was set to *p* < 0.05. All analyses were performed using R software (version 3.6.1, R Foundation for Statistical Computing, Vienna, Austria).

## Figures and Tables

**Figure 1 toxins-12-00715-f001:**
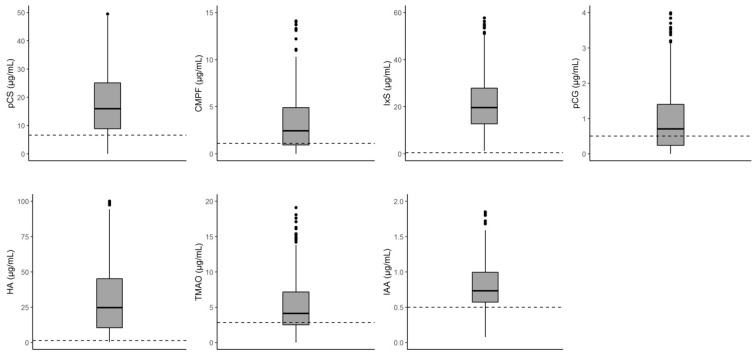
Distribution of UT concentrations at the time of transplantation The dashed lines indicates the corresponding reference mean UT concentrations in adults with normal kidney function [34,35]. CMPF, 3-carboxy-4-methyl-5-propyl-furanpropionic acid; HA, hippuric acid; IAA, indole-3-acetic acid; IxS, indoxylsulfate; pCG, p-cresylglucuronide; pCS, p-cresylsulfate; TMAO, trimethylamine-N-oxide.

**Figure 2 toxins-12-00715-f002:**
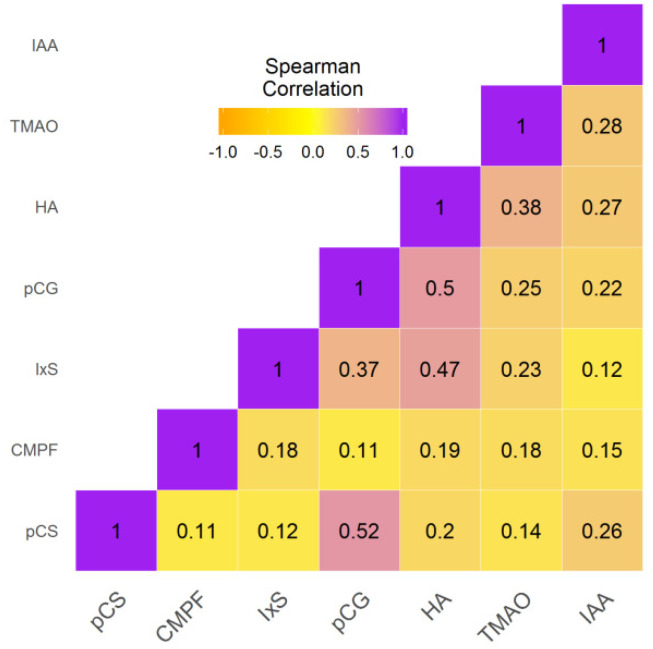
Correlation matrix for UT concentrations at baseline. CMPF, 3-carboxy-4-methyl-5-propyl-furanpropionic acid; HA, hippuric acid; IAA, indole-3-acetic acid; IxS, indoxylsulfate; pCG, p-cresylglucuronide; pCS, p-cresylsulfate; TMAO, trimethylamine-N-oxide.

**Figure 3 toxins-12-00715-f003:**
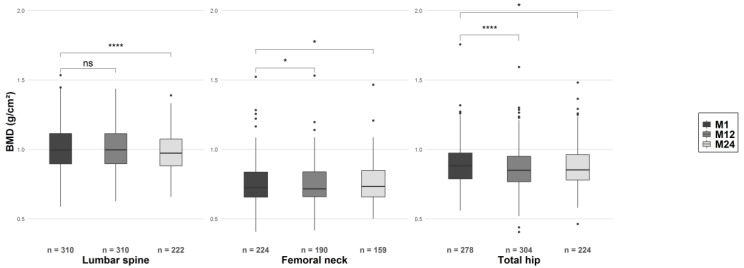
Change over time in BMD at the lumbar spine, femoral neck, and total hip, 12 and 24 months after transplantation. ns, nonsignificant; * *p* < 0.05, **** *p* < 0.001.

**Figure 4 toxins-12-00715-f004:**
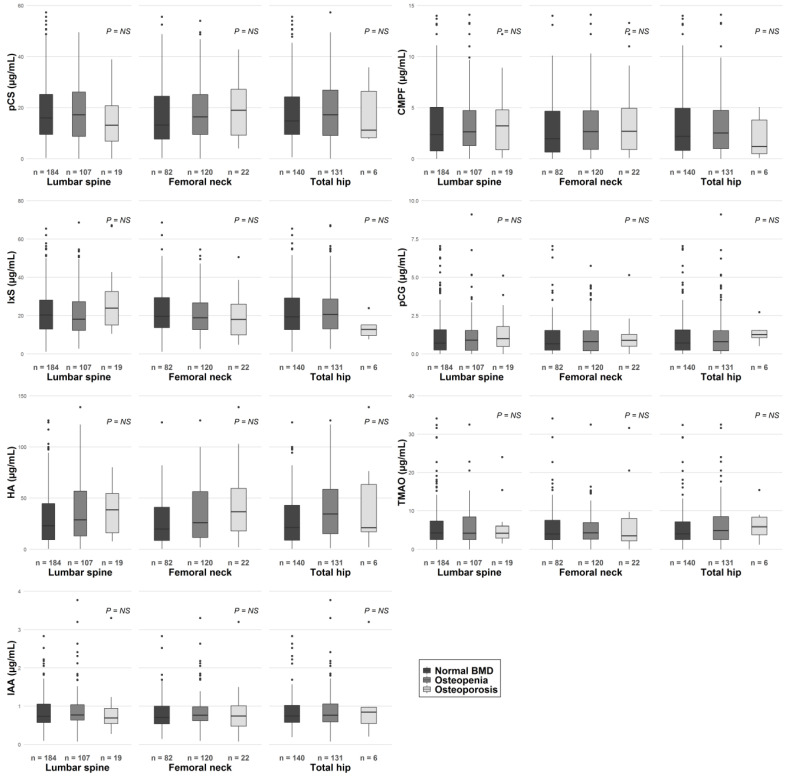
Distribution of UT concentrations upon transplantation, by osteoporosis status. CMPF, 3-carboxy-4-methyl-5-propyl-furanpropionic acid; HA, hippuric acid; IAA, indole-3-acetic acid; IxS, indoxylsulfate; pCG, p-cresylglucuronide; pCS, p-cresylsulfate; TMAO, trimethylamine-N-oxide.

**Table 1 toxins-12-00715-t001:** Description of the study population.

Characteristics of the Study Population	Population *n* = 310
Clinical risk factors for osteoporosis (M0)	
Recipient age (years), mean ± SD	51.1 ± 12.8
Female sex, *n* (%)	116 (37.4)
Ethnic group Caucasian, *n* (%) Black, *n* (%)	292 (94.2) 18 (5.8)
BMI (kg/m²), mean ± SD	26.2 ± 4.3
Thyroid disorders, *n* (%)	17 (5.5)
Prior osteoporotic fractures, *n* (%)	30 (9.7)
Family history of FUEF, *n* (%)	2 (0.6)
Diabetes mellitus, *n* (%)	50 (16.1)
Chronic inflammatory rheumatism, *n* (%)	4 (1.3)
Autoimmune diseases, *n* (%)	20 (6.5)
Primary HPT, *n* (%)	8 (2.6)
Secondary HPT, *n* (%)	266 (85.8)
Smoking Never, *n* (%) Current, *n* (%) Past, *n* (%)	159 (51.3) 88 (28.4) 63 (20.3)
Alcohol consumption, *n* (%)	22 (7.1)
Menopausal women, *n* = 116	51 (50.3)
Laboratory data	
Serum calcium (mg/L), mean ± SD	93.8 ± 8.0
Serum phosphate (mg/L), mean ± SD	45.3 ± 14.6
Serum 25(OH) vitamin D (ng/mL), mean ± SD	33.4 ± 16.2
Serum PTH (pg/mL), median (range)	334.0 (1.3–2646.0)
Serum bone alkaline phosphatases (µg/L), median (range)	12.3 (2.4–99.0)
Serum osteocalcin (ng/mL), median (range)	74.7 (2.0–2970.0)
Serum creatinine at M12 (mg/L), mean ± SD	16.2 ± 7.5
GFR at M12 (mL/min), mean ± SD	51.3 ± 20.2
Drugs influencing BMD	
Prior steroid intake, *n* (%)	55 (17.7)
Prior calcium intake, *n* (%)	84 (27.1)
Calcium intake during the study period, *n* (%)	106 (34.2)
Prior vitamin D intake ^1^, *n* (%)	154 (49.7)
Cholecalciferol, *n* (%)	113 (35.5)
Alfacalcidiol, *n* (%)	17 (5.5)
Calcifediol, *n* (%)	42 (13.5)
Vitamin D intake during the study period ^1^, *n* (%)	267 (86.1)
Cholecalciferol, *n* (%)	252 (81.3)
Alfacalcidiol, *n* (%)	32 (10.3)
Calcifediol, *n* (%)	43 (13.9)
Prior BP intake, *n* (%)	2 (0.6)
BP intake during the study period, *n* (%)	11 (3.5)
Etiology of chronic kidney disease	
Glomerulonephritis, *n* (%)	92 (29.7)
Hereditary disease, *n* (%)	58 (18.7)
Polycystic kidney disease, n *(%)*	55 (17.7)
Renal and urinary tract malformations, *n* (%)	29 (9.4)
Hypertensive kidney disease, *n* (%)	24 (7.7)
Diabetic kidney disease, *n* (%)	23 (7.4)
Interstitial nephritis, *n* (%)	13 (4.2)
Vascular nephropathy, *n* (%)	11 (3.5)
Indeterminate, *n* (%)	44 (14.2)
Other, *n* (%)	16 (5.2)
Time on hemodialysis (years), *median (range)*	2.5 (0–30.7)
Previous kidney transplant, *n* (%) 1, *n* (%) 2, *n* (%)	37 (11.9) 6 (1.9)
Prior cinacalcet intake, *n* (%)	76 (24.5)
Prior ESA intake, *n* (%)	75 (24.2)
Peak PRAs <20%, *n* (%) 20–80%, *n* (%) >80%, *n* (%)	224 (72.3) 59 (19.0) 27 (8.7)
DSAs, *n* (%) Previous, *n* (%) Current, *n* (%)	7 (2.3) 4 (1.3)
Positive crossmatch, *n* (%)	7 (2.3)
Donor Age (years), mean ± SD Female, *n* (%) BMI (kg/m²), mean ± SD Decreased donor, *n* (%) Last blood creatinine (mg/L), mean ± SD	51.9 ± 14.4 140 (45.2) 26.2 ± 5.7 282 (91.0) 80.7 ± 45.0
Ischemia times Cold (minutes), median (range) Warm (minutes), median (range)	806 (22–2036) 59 (4–99)
Induction therapy Basiliximab, *n* (%) Thymoglobulin, *n* (%) IVIg, *n* (%)	163 (52.6) 149 (48.1) 13 (4.2)
Maintenance therapy MMF + tacrolimus, *n* (%) MMF + cyclosporine, *n* (%) Tacrolimus + everolimus, *n* (%) MMF + everolimus, *n* (%) Tacrolimus + azathioprine, *n* (%)	208 (67.1) 87 (28.1) 13 (4.2) 1 (0.3) 1 (0.3)
Early steroid withdrawal, *n* (%)	41 (13.2)
Uremic toxin concentrations at transplantation	
pCS (mg/mL), med (IQR)	16.1 (9.02–25.60)
CMPF (mg/mL), med (IQR)	2.57 (0.97–5.08)
IxS (mg/mL), med (IQR)	19.80 (12.82–28.32)
pCG (mg/mL), med (IQR)	0.80 (0.25–1.58)
HA (mg/mL), med (IQR)	25.45 (10.90–50.15)
TMAO (mg/mL), med (IQR)	4.27 (2.54–7.78)
IAA (mg/mL), med (IQR)	0.75 (0.58–1.06)
Occurrence of fractures	
Within 12 months of transplantation, *n* (%)	4 (1.3)
Within 24 months of transplantation, *n* (%)	4 (1.3)

BMD, bone mineral density; BMI, body mass index; BP, bisphosphonate; CMPF, 3-carboxy-4-methyl-5-propyl-furanpropionic acid; DSA, donor-specific antibody; ESA, erythropoiesis-stimulating agent; FUEF, fracture of upper extremity of femur; GFR, glomerular filtration rate; HPT, hyperparathyroidism; HA, hippuric acid; IAA, indole-3-acetic acid; IQR, interquartile range; IVIg, intravenous immunoglobulin; IxS, indoxylsulfate; MMF, mycophenolate mofetil; pCG, p-cresylglucuronide; pCS, p-cresylsulfate; PRA, panel-reactive antibody; PTH, parathyroid hormone; TMAO, trimethylamine-N-oxide; ^1^ A given patient could have received cholecalciferol and one of its metabolites but not the two concomitantly.

**Table 2 toxins-12-00715-t002:** Correlations between UT concentrations and BMD at M1.

Site of BMD	pCS		CMPF		IxS		pCG		HA		TMAO		IAA	
	rho	*p*	rho	*p*	rho	*p*	rho	*p*	rho	*p*	rho	*p*	rho	*p*
Lumbar spine, *n =* 310	+0.02	0.782	+0.01	0.805	+0.04	0.477	−0.01	0.876	−0.06	0.300	+0.01	0.797	+0.03	0.656
Femoral neck, *n =* 145	+0.02	0.730	−0.05	0.448	+0.13	0.049	+0.03	0.701	−0.14	0.036	−0.00	0.975	−0.07	0.282
Total hip, *n* = 276	−0.01	0.913	+0.03	0.592	+0.13	0.037	+0.01	0.901	−0.08	0.190	−0.00	0.942	−0.02	0.738

BMD, bone mineral density; CMPF, 3-carboxy-4-methyl-5-propyl-furanpropionic acid; HA, hippuric acid; IAA, indole-3-acetic acid; IxS, indoxylsulfate; pCG, p-cresylglucuronide; pCS, p-cresylsulfate; rho, Spearman’s correlation coefficient; TMAO, trimethylamine-N-oxide.

**Table 3 toxins-12-00715-t003:** Comparison of UT concentrations in patients with BMD loss vs. patients with BMD gain 12 months after kidney transplantation.

UT Concentrations	Lumbar Spine			Femoral Neck			Total Hip		
	BMD Loss *n* = 154	BMD Gain *n* = 156	*p*	BMD Loss *n* = 89	BMD Gain *n* = 56	*p*	BMD Loss *n* = 197	BMD Gain *n* = 79	*p*
pCS (µg/mL)	16.2 (0.02–65.7)	15.9 (0.3–68.1)	0.857	13.9 (0.1–52.5)	15.4 (0.5–65.7)	0.556	16.0 (0.1–68.1)	16.0 (0.6–57.3)	0.964
CMPF (µg/mL)	3.0 (0.0–32.3)	2.0 (0.0–18.0)	0.005	2.9 (0.0–22.0)	2.2 (0.0–32.3)	0.070	2.5 (0.0–28.3)	2.4 (0.0–18.0)	0.921
IxS (µg/mL)	20.4 (1.2–101.0)	19.6 (1.8–67.2)	0.401	19.3 (1.8–68.6)	19.9 (1.3–54.5)	0.861	19.8 (1.2–67.2)	20.7 (3.7–101.0)	0.476
pCG (µg/mL)	0.89 (0.00–9.10)	0.71 (0.00–6.80)	0.368	0.72 (0.00–7.04)	0.66 (0.00–5.75)	0.488	0.80 (0.00–9.10)	0.69 (0.00–5.17)	0.786
HA (µg/mL)	25.6 (0.4–195.0)	25.7 (1.0–139.0)	0.842	28.0 (1.8–139.0)	25.7 (0.4–100.0)	0.932	24.0 (0.4–195.0)	30.7 (1.1–100.0)	0.471
TMAO (µg/mL)	4.4 (0.0–32.5)	4.2 (0.0–54.0)	0.577	4.2 (0.0–22.7)	4.1 (0.0–32.5)	0.534	4.3 (0.0–54.0)	4.5 (0.0–32.4)	0.433
IAA (µg/mL)	0.75 (0.08–7.61)	0.75 (0.10–5.28)	0.302	0.74 (0.14–7.61)	0.75 (0.24–5.89)	0.940	0.74 (0.19–7.61)	0.78 (0.08–5.89)	0.808

CMPF, 3-carboxy-4-methyl-5-propyl-furanpropionic acid; HA, hippuric acid; IAA, indole-3-acetic acid; IxS, indoxylsulfate; pCG, p-cresylglucuronide; pCS, p-cresylsulfate; TMAO, trimethylamine-N-oxide; UT, uremic toxin.

**Table 4 toxins-12-00715-t004:** Correlations between UT concentrations and BMD changes 12 and 24 months after transplantation.

Site of BMD		pCS		CMPF		IxS		pCG		HA		TMAO		IAA	
		rho	*p*	rho	*p*	rho	*p*	rho	*p*	rho	*p*	rho	*p*	rho	*p*
Lumbar spine	12 months after transplantation, *n =* 310	−0.02	0.747	−0.09	0.116	+0.02	0.736	−0.03	0.624	+0.02	0.666	−0.01	0.901	−0.03	0.569
	24 months after transplantation, *n =* 222	−0.00	0.967	−0.10	0.135	−0.03	0.662	+0.03	0.621	−0.03	0.614	+0.07	0.320	−0.06	0.368
Femoral neck	12 months after transplantation, *n =* 145	−0.10	0.229	+0.10	0.218	+0.04	0.654	−0.01	0.908	−0.03	0.760	+0.07	0.397	+0.01	0.869
	24 months after transplantation, *n =* 113	+0.03	0.747	−0.03	0.733	+0.11	0.244	+0.06	0.533	+0.13	0.173	+0.07	0.443	+0.05	0.576
Total hip	12 months after transplantation, *n =* 276	−0.05	0.389	−0.02	0.770	−0.02	0.763	−0.04	0.545	−0.07	0.257	−0.00	0.976	+0.07	0.238
	24 months after transplantation, *n =* 200	−0.05	0.509	−0.02	0.730	+0.04	0.570	+0.01	0.868	+0.02	0.743	+0.03	0.671	−0.01	0.944

BMD, bone mineral density; CMPF, 3-carboxy-4-methyl-5-propyl-furanpropionic acid; HA, hippuric acid; IAA, indole-3-acetic acid; IxS, indoxylsulfate; pCG, p-cresylglucuronide; pCS, p-cresylsulfate; rho, Spearman’s correlation coefficient; TMAO, trimethylamine-N-oxide.

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
