# Peer review of "Association between Uremic Toxin Concentrations and Bone Mineral Density after Kidney Transplantation"

_toxins, 2020, doi:10.3390/toxins12110715_

Round 1

Reviewer 1 Report

In the manuscript entitled: “Association between uremic toxin concentrations and bone mineral density after kidney transplantation”, the authors determine whether concentrations of UTs (trimethylamine-N-oxide, indoxylsulfate, p-cresylsulfate, p-cresylglucuronide, indole-3-acetic acid, hippuric acid, and 3-carboxy-4-methyl-5-propyl-furanpropionic acid) at the time of transplantation are correlated with BMD and predictive of BMD changes and the occurrence of fracture after kidney transplantation.

The authors concluded that uremic toxin concentrations upon transplantation were not associated with osteoporosis or BMD 1 month after transplantation or with BMD changes and the occurrence of osteoporotic fracture 1, 12 and 24 months after transplantation. Hence, UT concentrations at the time of kidney transplantation were not predictive markers of osteoporosis or fractures.

Major comments:

In general, the idea and innovation of this study, regards the analysis of toxin concentrations and bone mineral density is interesting, because the type of treatment of bone diseases is validated but further studies on this topic could be an innovative issue in this field and could be open a creative matter of debate in literature by adding new information. Moreover, there are few reports in the literature that studied this interesting topic with this kind of study design.

The study was well conducted by the authors; However, there are some concerns to revise that are described below.

The introduction section resumes the existing knowledge regarding the important factor linked with related bone disease.

However, as the importance of the topic, the reviewer strongly recommends, before a further re-evaluation of the manuscript, to update the literature through read, discuss and must cites in the references with great attention all of those recent interesting articles, that helps the authors to better introduce and discuss the role of periodontitis such as main disease linked with bone resorption in patients with CKD: 1) -           Isola G, Alibrandi A, Currò M, Matarese M, Ricca S, Matarese G, Ientile R, Kocher T. Evaluation of salivary and serum ADMA levels in patients with periodontal and cardiovascular disease as subclinical marker of cardiovascular risk. J Periodontol, 2020; 91:1076–1084. doi: 10.1002/JPER.19-0446. 2) -           Currò M, Matarese G, Isola G, Caccamo D, Ventura VP, Cornelius C, Lentini M, Cordasco G, Ientile R. Differential expression of transglutaminase genes in patients with chronic periodontitis. Oral Dis. 2014 Sep;20(6):616-23. doi: 10.1111/odi.12180.

The authors should be better specified, at the end of the introduction section, the rationale of the study. In the material and methods section, should better clarify patients selections and factors influencing BMD at M1 and changes in BMD at M12.

The discussion section appears well organized with the relevant paper that support the conclusions, even if the authors should better discuss the role of alveolar bone resorption and the quality of life in these patients. The conclusion should reinforce in light of the discussions.

In conclusion, I am sure that the authors are fine clinicians who achieve very nice results with their adopted protocol. However, this study, in my view does not in its current form satisfy a very high scientific requirement for publication in this journal and requests a revision before a futher re-evaluation of the manuscript.

Minor Comments:

Abstract:

  • Better formulate the abstract section by better describing the aim of the study

Introduction:

  • Please refer to major comments

Discussion

  • Please add a specific sentence that clarifies the results obtained in the first part of the discussion
  • Page 9 last paragraph: Please reorganize this paragraph that is not clear

Author Response

In the manuscript entitled: “Association between uremic toxin concentrations and bone mineral density after kidney transplantation”, the authors determine whether concentrations of UTs (trimethylamine-N-oxide, indoxylsulfate, p-cresylsulfate, p-cresylglucuronide, indole-3-acetic acid, hippuric acid, and 3-carboxy-4-methyl-5-propyl-furanpropionic acid) at the time of transplantation are correlated with BMD and predictive of BMD changes and the occurrence of fracture after kidney transplantation.

The authors concluded that uremic toxin concentrations upon transplantation were not associated with osteoporosis or BMD 1 month after transplantation or with BMD changes and the occurrence of osteoporotic fracture 1, 12 and 24 months after transplantation. Hence, UT concentrations at the time of kidney transplantation were not predictive markers of osteoporosis or fractures.

We thank the reviewer for his/her comments, which helped us to improve the manuscript.

Major comments:

In general, the idea and innovation of this study, regards the analysis of toxin concentrations and bone mineral density is interesting, because the type of treatment of bone diseases is validated but further studies on this topic could be an innovative issue in this field and could be open a creative matter of debate in literature by adding new information. Moreover, there are few reports in the literature that studied this interesting topic with this kind of study design.

The study was well conducted by the authors; however, there are some concerns to revise that are described below.

The introduction section resumes the existing knowledge regarding the important factor linked with related bone disease.

However, as the importance of the topic, the reviewer strongly recommends, before a further re-evaluation of the manuscript, to update the literature through read, discuss and must cites in the references with great attention all of those recent interesting articles, that helps the authors to better introduce and discuss the role of periodontitis such as main disease linked with bone resorption in patients with CKD: 1) -           Isola G, Alibrandi A, Currò M, Matarese M, Ricca S, Matarese G, Ientile R, Kocher T. Evaluation of salivary and serum ADMA levels in patients with periodontal and cardiovascular disease as subclinical marker of cardiovascular risk. J Periodontol, 2020; 91:1076–1084. doi: 10.1002/JPER.19-0446. 2) -           Currò M, Matarese G, Isola G, Caccamo D, Ventura VP, Cornelius C, Lentini M, Cordasco G, Ientile R. Differential expression of transglutaminase genes in patients with chronic periodontitis. Oral Dis. 2014 Sep;20(6):616-23. doi: 10.1111/odi.12180.

Thanks for this comment, which enabled us to improve the rationale and the Discussion.

We have now added new paragraphs to introduce (page 3, lines 77-80) and then discuss (page 14-15, lines 280-304) the role of periodontitis as the main disease to bone resorption in patients with CKD. We now also cite the suggested articles (page 3, line 80).

The authors should be better specified, at the end of the introduction section, the rationale of the study.

The study rationale is now explained in more detail (pages 3-4, lines 99-108).

In the material and methods section, should better clarify patients selections and factors influencing BMD at M1 and changes in BMD at M12.

The Material and Methods section has been completed accordingly (page 17, lines 393-394; page 17, lines 408-412; pages 17-18, lines 422-438).

The discussion section appears well organized with the relevant paper that support the conclusions, even if the authors should better discuss the role of alveolar bone resorption and the quality of life in these patients. The conclusion should reinforce in light of the discussions.

The role of alveolar bone resorption and quality of life in these patients are now discussed (page 15, lines 296-304). The conclusion has been modified accordingly (pages 16-17, lines 386-388).

Minor Comments:

 Abstract:

Better formulate the abstract section by better describing the aim of the study

The abstract section has been modified accordingly (page 1, lines 33-39).

Introduction:

Please refer to major comments

The Introduction has been modified accordingly (page 3, lines 77-80).

Discussion

Please add a specific sentence that clarifies the results obtained in the first part of the discussion.

The first part of the Discussion has now been clarified (page 14, lines 249-255).

Page 9 last paragraph: Please reorganize this paragraph that is not clear

This paragraph has now been reorganized (pages 9-10, lines 176-188).

Reviewer 2 Report

This study investigated the association between uremic toxins and bone mineral density in participants with kidney transplantation. Some research suggestions were advised.

  1. Page 1, line 32-33. The sentence of “Skeletal resistance to the action of PTH leads to adynamic bone disease” is misleading. Please revise. Please check the definition of adynamic bone disease.

  1. The study design was single pre-transplantation uremic toxins level to predict three times post-transplantation mineral bone status. The rationale should be more precise. As expected, the uremic toxin level dramatically decreased after kidney transplantation. How to link the single protein-bound uremic toxin level to post-transplantation mineral bone status? In addition, the abnormality of calcium, phosphate, and parathyroid hormone, which are all important parameters for the mineral bone disorder, improve rapidly after kidney transplantation. How to evaluate these confounding factors should be considered more in the current study?

  1. Page 11, line 230. Barreto et al.’s clinical study [52]…. I didn’t see reference 52. Is there something missing here?

  1. In the discussion section (page 11, line 234), the sentence “One can hypothesize that UTs diffuse into bone (which might explain these discrepancies), but no data on this topic are available” is confusing. Please specify this section and discuss more the discrepancy between animal study and human study about uremic toxins and bone parameters.

  1. How to explain a positive correlation between IxS and femoral neck and total hip BMD at M1? Please specify this part.

  1. Please provide the information on the pre-transplant condition. Are patients at CKD status or receiving dialysis?

Author Response

This study investigated the association between uremic toxins and bone mineral density in participants with kidney transplantation. Some research suggestions were advised.

We thank the reviewer for his/her comments, which helped us to improve the manuscript.

1.Page 1, line 32-33. The sentence of “Skeletal resistance to the action of PTH leads to adynamic bone disease” is misleading. Please revise. Please check the definition of adynamic bone disease.

The misleading sentence has been modified and the definition of adynamic bone disease has been checked (page 3, lines 65-71)

2.The study design was single pre-transplantation uremic toxins level to predict three times post-transplantation mineral bone status. The rationale should be more precise. As expected, the uremic toxin level dramatically decreased after kidney transplantation. How to link the single protein-bound uremic toxin level to post-transplantation mineral bone status? In addition, the abnormality of calcium, phosphate, and parathyroid hormone, which are all important parameters for the mineral bone disorder, improve rapidly after kidney transplantation. How to evaluate these confounding factors should be considered more in the current study?

The rationale has now been clarified (pages 3-4, lines 99-108).

A multiple linear regression has been performed in order to consider confounding factors for mineral bone disorders (Supplementary Table S9, page 9). The Methods section has been modified accordingly (page 19, lines 477-480). After adjustment for confounding factors, IxS concentrations upon transplantation were still positively correlated with BMD changes for the femoral neck at M1 (paragraphs have been added to the Results section on page 9, lines 179-183, to the Discussion on page 14, lines 249-251 and page 16, line 349 and to the abstract page 1, lines 42-44).

Moreover, after adjustment, BMD at M1 was (i) positively correlated with BMI at all measurement sites (lumbar spine, p<0.001; femoral neck, p=0.001; total hip, p<0.001), (ii) negatively correlated with the serum calcium level for the lumbar spine (p=0.006) and the total hip (p=0.005), and (iii) negatively correlated with the serum PTH level for the lumbar spine (p=0.014). BMD at M1 was significantly higher in men than in women for all measurement sites (lumbar spine, p=0.002; femoral neck, p=0.004; total hip, p<0.001). This paragraph has been added to the Results section on page 13 lines 216-221.

3.Page 11, line 230. Barreto et al.’s clinical study [52]…. I didn’t see reference 52. Is there something missing here?

This oversight has been rectified (page 15, line 336; page 24, lines 713-716).

4.In the discussion section (page 11, line 234), the sentence “One can hypothesize that UTs diffuse into bone (which might explain these discrepancies), but no data on this topic are available” is confusing. Please specify this section and discuss more the discrepancy between animal study and human study about uremic toxins and bone parameters.

The discrepancy between animal studies and human studies of uremic toxins and bone parameters is now discussed (page 15, lines 332-342).

5.How to explain a positive correlation between IxS and femoral neck and total hip BMD at M1? Please specify this part.

This positive correlation between IxS and femoral neck and total hip BMD at M1 corroborated the previous human study of uremic toxins and bone parameters. This was discussed on page 16, lines 347-353.

6.Please provide the information on the pre-transplant condition. Are patients at CKD status or receiving dialysis?

The study population consisted of dialysis patients. This point has been added to the patients and methods section on page 17, line 393.

Round 2

Reviewer 1 Report

The authors have well addressed to all reviewer's comments. I suggest the acceptance of this interesting manuscript.

Reviewer 2 Report

All comments had been revised and illustrated. I have no further suggestions.